# Competition between Invasive Ruffe (*Gymnocephalus cernua*) and Native Yellow Perch (*Perca flavescens*) in Experimental Mesocosms

**Raymond M. Newman** [1,*], **Fred G. Henson** [1,2] **and Carl Richards** [3]

1. Department of Fisheries, Wildlife and Conservation Biology, University of Minnesota, Saint Paul, MN 55108, USA; fred.henson@dec.ny.gov
2. New York State Department of Environmental Conservation, Division of Fish and Wildlife, Albany, NY 12233-4753, USA
3. Center for Water and the Environment, Natural Resources Research Institute, 5013 Miller Trunk Highway, Duluth, MN 55811, USA; crich555@gmail.com
* Correspondence: RNewman@umn.edu

**Abstract:** Ruffe (*Gymnocephalus cernua*) were introduced to North America from Europe in the mid-1980s and based on similar diets and habit use may compete with yellow perch (*Perca flavescens*). To examine competitive interactions between invasive ruffe and native yellow perch, individually marked perch and ruffe were placed in mesocosms in a small lake. Mesocosms allowed fish to interact and feed on the natural prey populations enclosed. In the first experiment, four treatments were assessed: 28 perch, 14 perch + 14 ruffe, 14 perch, and 7 perch + 7 ruffe. Yellow perch growth was significantly lower in the presence of ruffe (ANOVA, $p = 0.005$) than in treatments containing only perch. In a second experiment, an increasing density of one species was superimposed upon a constant density of the other in parallel treatment series. Growth rates of both ruffe and perch declined when ruffe density was increased (*t* test, $p = 0.006$). However, neither ruffe nor perch growth was affected by increasing perch density. Total stomach content mass of perch was significantly decreased by ruffe in both years ($p < 0.02$), but no effects of ruffe on the composition of perch diets were observed. Ruffe growth and food consumption was greater than that of perch for both experiments. Ruffe can outcompete yellow perch when both species depend on a limited benthic food resource. Thus there is reason for concern for the ecological effects of ruffe if they expand their range into Lake Erie or North American inland lakes that contain yellow perch.

**Keywords:** interference competition; exploitative competition; invasive species; ruffe; yellow perch; growth; diet

## 1. Introduction

The ruffe (*Gymnocephalus cernua*) is a percid fish, native to southern England, northeastern France, and central Europe eastward through Siberia that was introduced to North America in the mid 1980s, likely via ballast water from ships departing the northern Elbe River or eastern North Sea [1–3]. Ruffe were first collected from the Duluth-Superior Harbor in 1986 and became the most abundant fish in bottom trawl samples by 1991 [4]. While ruffe flourished in the harbor, several native fish populations, including yellow perch (*Perca flavescens*), declined [4,5]. Ruffe subsequently expanded along the North Shore of Lake Superior to Thunder Bay, Ontario, and along the near-shore and tributary waters of southern Lake Superior to Lake Huron [3,6]. Populations developed in Green Bay and Little Bay de Noc in Lake Michigan and in Thunder Bay and the Cheboygan River, Lake Huron, likely the result of inter-lake shipping ballast and natural dispersal [3,7]. Despite concern for their expansion

into the southern Great Lakes, ruffe have not been detected in southern Lake Michigan, Lake St. Clair or Lake Erie [7] nor have they been detected in any inland waters not tributary to Lakes Superior, Michigan or Huron [3].

In its native range in Europe and Asia, the ruffe is, at best, of marginal value as a fisheries resource and is widely considered a nuisance species [8]. Concern in North America is for potential negative effects on desired game and forage fishes. Within four years of their discovery in the Duluth Superior Harbor, ruffe became the most abundant fish and declines of other forage fish such as perch and trout perch were noted [4]. The observed declines in native fish abundance since the introduction of ruffe were more likely the consequence of natural population dynamics rather than an effect of ruffe, however, in the case of yellow perch, ruffe were partially responsible for fluctuations in year class strength [4]. Ruffe feed heavily on benthic invertebrates [9–12] and diet overlap suggests that exploitative competition between ruffe, yellow perch, and other native benthivores may occur [11,13–15]. Ruffe are particularly adapted to low light benthic habitats [11,16–18] and will likely be most successful in these conditions

Several laboratory studies have examined the potential for competition between ruffe and yellow perch. Ruffe and yellow perch were found to have similar prey preferences in the laboratory [15], and Fullerton et al. [19] found that perch and ruffe growth decreased with ruffe density in laboratory competition experiments for food between ruffe and yellow perch. However, they noted that overall fish density was more important to fish growth than the presence of ruffe. Savino and Kolar [13] concluded that ruffe compete with yellow perch, but the outcome depended on the situation. Ruffe were more efficient with unlimited food, but perch appeared to do better in food limited situations. However, ruffe were more aggressive than perch and may have an advantage over perch via interference competition. Savino and Kostich [20] noted intraspecific interference competition by ruffe and suggested ruffe will do best at intermediate densities. Fullerton and Lamberti [16] assessed both habitat use and feeding efficiency of ruffe and yellow perch and found no evidence of competition for habitat, but that within shared habitats competition for food may occur when food is limiting. The lower growth and conversion efficiency of ruffe suggests that ruffe will place a greater demand on benthic food resources than an equivalent biomass of perch [21] and thus could increase the potential for competition. Perch may be more effective in macrophyte habitats and ruffe appear also to be less adapted to compete with round gobies [22], particularly at low food densities and sand or macrophyte habitats.

These studies were conducted in aquaria and small tanks (100–280 L), although Bergman and Greenberg [23] documented competition between ruffe and European perch (*Perca fluviatilis*) in larger experimental mesocosms. They showed a decline in European perch growth rate and a diet shift by perch from benthic macroinvertebrates to microcrustaceans when ruffe density was increased. Given the high value of the yellow perch to Great Lakes fisheries [24], the potential for ruffe to invade substantial portions of yellow perch's North American range [24,25], and mixed evidence of competition between these two species, a controlled field experiment, to rigorously test the hypothesis that ruffe will decrease the fitness of yellow perch, was desired. A better assessment of ruffe's competitive potential will further inform modeling efforts and risk assessments that are being regularly updated to predict impacts and concern for invasive species (e.g., [25–27]).

Measuring competition between organisms and understanding its role in communities has been a longstanding challenge to ecologists. To demonstrate that competition exists between two species, one must show that a limited resource is utilized by both species and that the fitness of one species is decreased in the presence of the other [28–30]. The manipulation of the abundances of putative competitors in a carefully controlled field experiment has become a widely accepted means of investigating competition [28]. In previous larger scale tests of competition Bergman and Greenberg [23] manipulated ruffe density to demonstrate competition with European perch and Hanson and Leggett [29,30] manipulated pumpkinseed (*Lepomis gibbosus*) and yellow perch density to assess competition between these species. Hanson and Leggett [29] compared mixed treatments equal

in total biomass to single species treatments, whereas Bergman and Greenberg [23] superimposed an increasing density of ruffe on a constant density of perch. Bergman and Greenberg [23] criticized designs of the type used by Hanson and Leggett [29,30] because the results are highly dependent on the densities chosen and because they do not lead to the development of a density response curve. However, the design of Bergman and Greenberg [23] did not allow them to determine which species was the superior competitor or control for the effect of total fish density. Recognizing the advantages and shortcomings of both approaches, we conducted experiments using designs of both types. We also examined the effect of different densities of perch superimposed on a constant density of ruffe to determine whether perch could affect ruffe fitness in addition to assessing the effect of ruffe on yellow perch. This multifaceted approach generated a more comprehensive picture of competition between two freshwater fishes than would have been possible with a more limited set of experiments.

## 2. Results

### 2.1. Growth

In the ruffe and fish density experiment (1996), perch consistently lost mass except in the absence of ruffe at low density (14 perch + 0 ruffe) (Figure 1). In contrast, ruffe at both densities displayed positive growth (Figure 2). The degree of perch mass loss was marginally greater at higher overall fish density, whether quantified as a treatment factor (low vs. high, ANOVA, $p = 0.057$) or as a covariate (measured treatment biomass, ANCOVA, $p = 0.050$). At the same overall fish density, perch mass loss was significantly greater in the presence of ruffe than in perch only treatments (ANOVA, $p = 0.005$), and there was also a significant interaction between overall fish density and the presence or absence of ruffe (ANOVA, $p = 0.041$). There was no significant effect of time (ANOVA, $p = 0.44$), block ($p = 0.20$), or interactions between other factors (time and overall fish biomass; all $p \geq 0.44$). The results were similar when gross change in mass was analyzed, except that there were no significant interactions and the effects of density (ANOVA, $p = 0.018$) and time (ANOVA, $p = 0.002$) were stronger. Perch growth became significantly more negative as the experiment progressed. There was no main effect of fish density on the growth of ruffe (ANOVA, $p = 0.320$), but a significant interaction between fish density and time (ANOVA, $p = 0.033$); fish density did affect ruffe growth in the middle of the experiment.

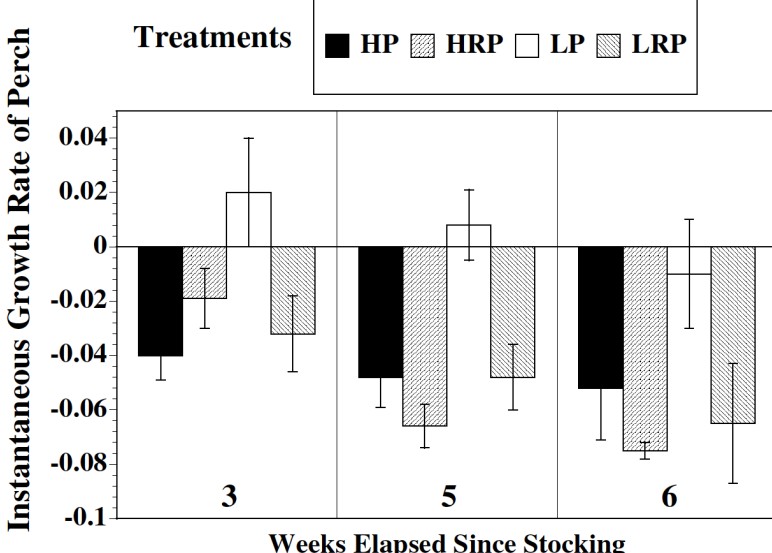

**Figure 1.** Mean instantaneous growth rate (plus or minus one standard error) by treatment, of yellow perch after three, five, and six weeks in the mesocosms. There were four replicates of each treatment except LRP ($n = 3$). HP = 28 perch, HRP = 14 perch + 14 ruffe, LP = 14 perch, and LRP = 7 perch + 7 ruffe.

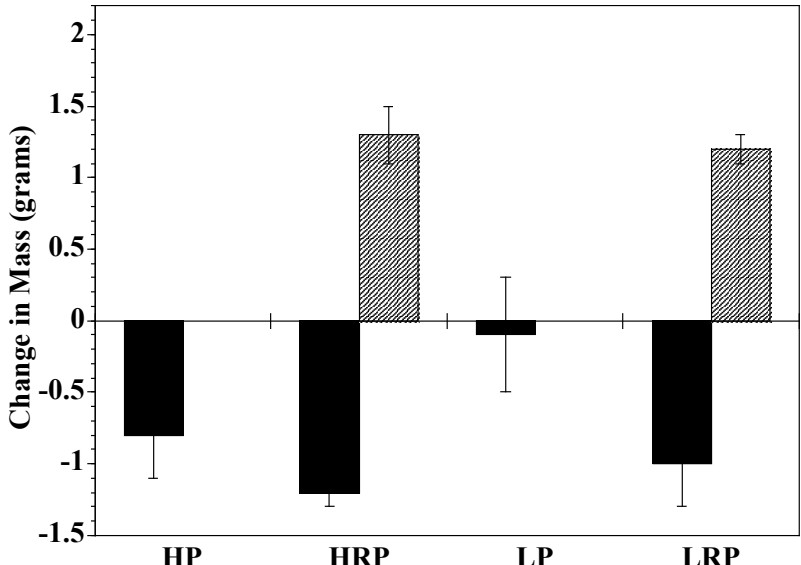

**Figure 2.** Gross change in mean individual mass (grams) of yellow perch (solid columns) and ruffe (striped columns) by treatment (plus or minus one standard error) during the six-week ruffe and fish density experiment. There are four replicates of each treatment except LRP (*n* = 3). HP = 28 perch, HRP = 14 perch + 14 ruffe, LP = 14 perch, and LRP = 7 perch + 7 ruffe.

As in the first experiment, ruffe growth was generally greater than perch growth in the density gradient experiment (Figures 3 and 4). Growth rate of all fish decreased as the number of ruffe increased (ANOVA, *p* = 0.043) (Figure 3). The two trials (August–September and September–October) were not significantly different (ANOVA, *p* = 0.110) and, because there was no significant interaction between ruffe density and species, the decrease in growth with increasing ruffe density was the same for perch and ruffe. Growth rate of perch and ruffe decreased at −0.037% * $d^{-1}$ * $ruffe^{-1}$. This slope was significantly different from zero (*t* test, *p* = 0.006) and was not significantly different from a linear decrease (ANOVA, *p* = 0.552). The addition of perch to a constant density of ruffe did not have a significant effect on either ruffe or perch growth rate (all *p* > 0.1, Figure 4). There was a significant difference between the mean growth of ruffe and perch (ANOVA, *p* < 0.001); ruffe grew faster than perch, who lost mass.

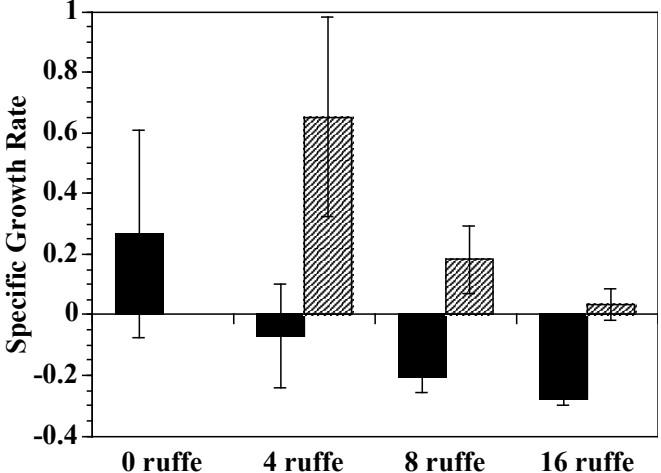

**Figure 3.** The effect of increasing ruffe density on specific growth rates of ruffe (striped columns) and yellow perch (solid columns) over the two five-week trials of the fish density gradient experiment (August–October 1997). Treatments consist of four replicates each of 0, 4, 8, and 16 ruffe added to 8 yellow perch. Treatment means are shown plus or minus one standard error.

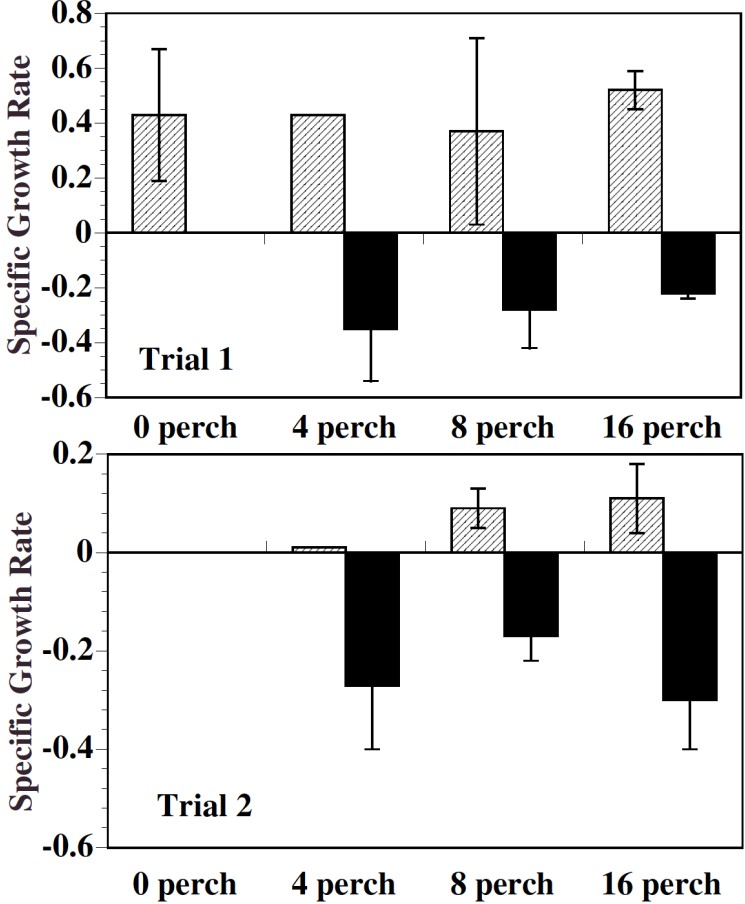

**Figure 4.** The effect of increasing yellow perch density on specific growth rates of ruffe (striped columns) and yellow perch (solid columns) over the two five-week trials of fish density gradient experiment (August–October 1997). Trials consist of two replicates each of 0, 4, 8, and 16 yellow perch added to 8 ruffe. Trial means are shown plus or minus one standard error. First and second trials are shown separately due to the effect of missing observations (8 ruffe + 0 perch) in second set.

*2.2. Diet*

In ruffe and fish density experiment, the mean mass of ruffe stomach contents was approximately three times greater than the mean mass of perch stomach contents across treatments. The mean mass of perch stomach contents was not significantly affected by any of the factors that affected perch growth in the repeated measures split-plot ANOVA including presence or absence of ruffe when all dates were considered. However, when diet data from the final (24 October 1996) sample were analyzed separately, mean stomach content mass of perch was significantly reduced in the presence of ruffe (ANOVA, $p = 0.009$; Figure 5). In this analysis, there was also a significant effect of block (ANOVA, $p = 0.024$), but no effect of overall fish density. There was no effect of ruffe on the proportion of microcrustaceans in the perch stomachs; Cladocera and Copepoda composed about 33% of perch diet but only 11% of ruffe diet.

Analysis of the gradient experiment diet data was hindered by low fish recovery from the second trial at the end of the experiment that was due in part to predation by an otter that was first observed in Perch Lake during the final week of the experiment. Therefore, we restricted diet analysis to the first two replicates. There was a significant negative effect of ruffe on the total mass of prey in perch stomachs (*t* test, $p = 0.014$), but no significant effect of ruffe on ruffe stomach content mass (Figure 6). Perch density did not affect the stomach content mass of either species. As in the first experiment, there was no significant effect of ruffe on the proportion of zooplankton in perch diet. Cladocera and Copepoda composed about 12% of perch diet and <1% of ruffe diet.

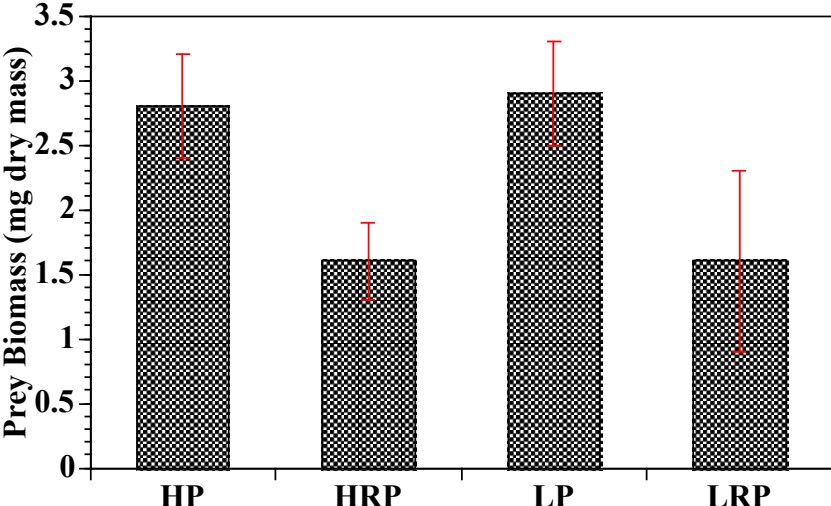

**Figure 5.** Estimated mass of individual yellow perch stomach contents collected at the conclusion of the ruffe and fish density experiment 1 (24 October 1996). Treatment means are shown plus or minus one standard error. There are four replicates of each treatment except LRP (*n* = 3). HP = 28 perch, HRP = 14 perch + 14 ruffe, LP = 14 perch, and LRP = 7 perch + 7 ruffe.

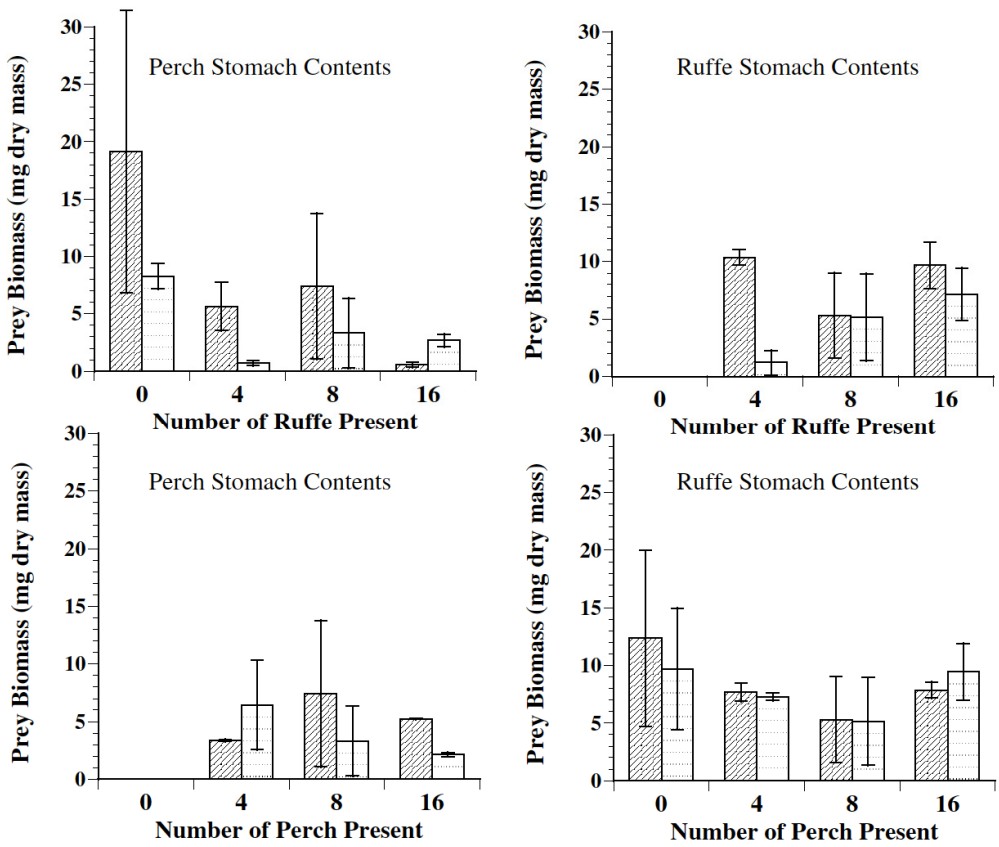

**Figure 6.** Estimated stomach content mass of yellow perch with increasing ruffe density (**top left**), ruffe with increasing ruffe density (**top right**), ruffe with increasing yellow perch density (**lower right**), and yellow perch with increasing yellow perch density (**lower left**) (experiment 2, 11 August–15 September). Stomach content mass is shown after 2.5 weeks (cross-hatching) and 5 weeks (horizontal stripes). Treatment means are shown plus or minus one standard error. There are two replicates of each treatment. There was a significant effect of ruffe on perch stomach content mass (*p* = 0.014), but no effect of ruffe on ruffe stomach content mass or of perch on either ruffe or perch stomach content mass (all *p* >0.1).

### 2.3. Food Consumption

Yellow perch daily individual food consumption ranged from 0.14–0.60 g (Tables 1 and 2). Ruffe daily individual food consumption ranged from 0.18–0.93 g. Food consumption by ruffe was generally twice that of perch sharing the same enclosures even though ruffe were generally smaller. In the ruffe and fish density experiment, perch daily ration was reduced in the presence of ruffe (ANOVA, $p = 0.03$) and in the gradient experiment, perch and ruffe daily ration declined with increasing ruffe density (both $p = 0.01$), but no declines in ruffe or perch daily ration were evident with perch density ($p > 0.7$) or total density ($p > 0.08$). Total daily consumption (ruffe and perch combined) increased with both total fish density ($p = 0.01$) and presence of ruffe ($p = 0.04$) in the first experiment. In the gradient experiment, total consumption increased linearly with total fish density and with yellow perch density (ANCOVA, both $p < 0.005$), but not with increasing ruffe density (ANCOVA, $p = 0.14$). The relationship was clearly asymptotic with increasing ruffe density; ln (total consumption) increased significantly with increasing ruffe density (ANCOVA, $p = 0.017$) and the fit was further improved with square root of ruffe density. The fit of the perch density or total density relationship was not improved with a logarithmic transformation of total consumption. Thus, increasing density of perch resulted in a direct increase in consumption, but increasing density of ruffe beyond about 12 ruffe per mesocosm resulted in little increase in total consumption, suggesting either severe intraspecific competition or severe food limitation.

**Table 1.** Estimates of average daily food consumption by ruffe and yellow perch during the first (1996) experiment, based on bioenergetics modeling. Daily ration (DR) is reported as percent wet mass and daily individual (IC) and total (TC) consumption (means for each treatment) are reported in grams wet mass per day. Standard errors (SE) are based on four replicates (mesocosms) of each treatment, except 7 Perch and 7 Ruffe, where $n = 3$.

| Treatment | 28 Perch | 14 Perch 14 Ruffe | 14 Perch | 7 Perch 7 Ruffe |
|---|---|---|---|---|
| Perch DR % | 2.09 (0.12) | 1.95 (0.14) | 2.35 (0.15) | 1.69 (0.12) |
| Perch IC (g) | 0.32 (0.02) | 0.30 (0.02) | 0.39 (0.03) | 0.27 (0.02) |
| Perch TC (g) | 9.02 (0.57) | 4.21 (0.28) | 5.41 (0.37) | 1.87 (0.11) |
| Ruffe DR % | | 6.05 (0.56) | | 5.47 (0.36) |
| Ruffe IC (g) | | 0.63 (0.05) | | 0.55 (0.06) |
| Ruffe TC (g) | | 8.77 (0.68) | | 3.88 (0.43) |
| Ruffe + Perch TC (g) | 9.02 (0.57) | 12.98 (0.71) | 5.41 (0.37) | 5.75 (0.51) |

**Table 2.** Estimates of average daily food consumption (mean and 1SE) by ruffe (R) and yellow perch (P) during the second (1997) experiment, based on bioenergetics modeling. Daily ration (DR) is reported as percent wet mass and daily individual (IC) and total (TC) consumption (means for each treatment) are reported in grams wet mass per day. Standard errors (SE) are based on two replicates (mesocosms) of each treatment except for those cases with no SE (-; $n = 1$) and 8R8P in September to October ($n = 4$).

| | 11 August–15 September | | | | | | |
|---|---|---|---|---|---|---|---|
| Treatment | 0R8P | 4R8P | 8R8P | 16R8P | 8R0P | 8R4P | 8R16P |
| Ruffe DR % | | 12.44 | 7.70 | 4.33 | 8.36 | 8.36 | 9.28 |
| SE | | 5.00 | 3.42 | 0.87 | 2.45 | 0.00 | 0.71 |
| Ruffe IC (g) | | 0.93 | 0.66 | 0.33 | 0.64 | 0.68 | 0.84 |
| SE | | 0.41 | 0.32 | 0.08 | 0.18 | 0.02 | 0.05 |
| Ruffe TC (g) | | 3.71 | 5.29 | 5.35 | 5.14 | 5.47 | 6.73 |
| SE | | 1.66 | 2.53 | 1.30 | 1.43 | 0.18 | 0.39 |
| Perch DR % | 5.23 | 3.99 | 3.17 | 3.13 | | 2.87 | 3.23 |
| SE | 0.98 | 0.86 | 0.23 | 0.16 | | 0.19 | 0.03 |
| Perch IC (g) | 0.60 | 0.29 | 0.34 | 0.33 | | 0.38 | 0.37 |
| SE | 0.04 | 0.04 | 0.08 | 0.07 | | 0.17 | 0.01 |
| Perch TC (g) | 4.77 | 2.35 | 2.69 | 2.62 | | 1.54 | 5.88 |
| SE | 0.31 | 0.30 | 0.65 | 0.58 | | 0.66 | 0.21 |
| Total C (g) | 4.77 | 6.06 | 7.98 | 7.97 | 5.14 | 7.00 | 12.61 |
| SE | 0.31 | 1.95 | 3.18 | 1.89 | 1.43 | 0.48 | 0.18 |

**Table 2.** *Cont.*

| | | | 22 September–28 October | | | | |
|---|---|---|---|---|---|---|---|
| Treatment | 0R8P | 4R8P | 8R8P | 16R8P | 8R0P | 8R4P | 8R16P |
| Ruffe DR % | | 5.43 | 3.28 | 2.67 | | 2.47 | 3.44 |
| SE | | - | 0.42 | - | | 0.00 | 0.66 |
| Ruffe IC (g) | | 0.54 | 0.25 | 0.18 | | 0.19 | 0.25 |
| SE | | - | 0.04 | - | | 0.02 | 0.06 |
| Ruffe TC (g) | | 2.17 | 2.03 | 2.95 | | 1.50 | 1.97 |
| SE | | - | 0.32 | - | | 0.16 | 0.46 |
| Perch DR % | 1.59 | 2.08 | 1.84 | 1.61 | | 1.50 | 1.53 |
| SE | - | - | 0.12 | 0.11 | | 0.33 | 0.23 |
| Perch IC (g) | 0.14 | 0.18 | 0.18 | 0.16 | | 0.18 | 0.15 |
| SE | - | - | 0.01 | 0.01 | | 0.04 | 0.02 |
| Perch TC (g) | 1.15 | 1.45 | 1.46 | 1.30 | | 0.72 | 2.45 |
| SE | - | - | 0.12 | 0.04 | | 0.15 | 0.39 |
| Total C (g) | 1.15 | 3.62 | 3.49 | 4.25 | | 2.22 | 4.42 |
| SE | - | - | 0.28 | - | | 0.01 | 0.85 |

## 3. Discussion

### 3.1. Growth

Competitive interactions are one vehicle for adverse effects of invasive species [31,32], however, the relative importance interspecific completion is unclear, and results of small-scale experiments may not translate to larger scales [33,34]. Relatively few studies have provided experimental evidence of competition between fishes at larger scales. The consistent negative effect of the presence of ruffe on the growth of yellow perch observed in both years of this study provides strong evidence that ruffe compete with yellow perch and that the ruffe is the superior competitor. Whereas the experimental design of Bergman and Greenberg [23] could not identify which species was the superior competitor, our experiments answered that important question. Ruffe was the superior competitor because perch growth in ruffe and the fish density experiment were depressed significantly more by ruffe than by an equivalent biomass of perch and, in the gradient experiment, growth of perch was reduced with increased ruffe density. Mean ruffe mass was less than mean perch mass, so these results may be considered somewhat conservative.

The marginal effect of fish density observed in the first experiment demonstrates intraspecific competition because perch growth was lower at the high-density perch-only treatment than at the low-density perch-only treatment. There was not, however, a similar negative effect of increasing perch density on perch growth in the gradient experiment. One plausible explanation is that because eight ruffe were present in all treatments, intraspecific competition among perch was masked by the stronger effect of interspecific competition with ruffe.

Perch density had no effect on growth of ruffe (Figure 4), suggesting that a dense, established perch population will not deter invading ruffe. However, the results are somewhat less clear concerning intraspecific competition among ruffe. In the gradient experiment, ruffe growth declined at the same rate as perch growth with increasing ruffe density, suggesting intraspecific competition. This observation is in accord with the results of Bergman and Greenberg [23] and Fullerton et al. [19], but the lack of an overall significant difference in ruffe growth between the high and low density mixed species treatments in the first experiment is contradictory. Neither ruffe nor perch growth or ration were significantly different between the high and low density treatments when ruffe were present. The gradient experiment, in combination with the observations of others [19,23], suggests strong intraspecific competition among ruffe; however, our overall results suggest that perch are more consistently affected by competition from ruffe. The degree of intraspecific competition among ruffe is important because, given the lack of a perch effect on ruffe growth, ruffe densities in the wild may ultimately be determined by intraspecific competition rather than by competition with other species. This supports the idea that ruffe should grow best at intermediate densities [20]. There was early evidence that ruffe growth

declined as their density increased in the Saint Louis River Harbor [5], but formal analyses of growth related to population density have not been reported for ruffe in North America. Although lengths have been reported along with CPUE (e.g., [35]), aging data, length at age, or growth rates are lacking in population assessments (e.g., [35,36]); these data for ruffe and their competitors would be useful to determine the effects of ruffe in invaded environments.

The negative growth rates of perch in many of the treatments indicates that conditions were not optimal for perch growth. Ruffe and perch were both collected from the wild and the larger perch may have been more stressed, although we held fish for a week before stocking to reduce use of stressed fish. Perch consumed prey throughout the experiments and did grow at low density in the absence of ruffe. The lower water clarity of Perch Lake and the low abundance of larger preferred zooplankton may have also contributed to their lack of positive growth. Finally, the larger size of perch may have resulted in them exceeding the carrying capacity of the mesocosms at total densities above 7 perch (the density at which perch growth was positive). Nonetheless, the consistent reduction of growth with increasing perch and ruffe density suggest that the effects of ruffe and our interpretation of competitive interactions are relevant to field conditions.

### 3.2. Diet and Consumption

Perch across all treatments were primarily benthivorous, but included substantial proportions of Copepoda and Cladocera in their diet (33% in 1996; 11% in 1997). However, in contrast to the results of similar competition experiments involving yellow perch or European perch [23,29], we found no evidence that perch ate a higher proportion of microcrustaceans in the presence of a superior benthic competitor or in response to increasing density of the superior competitor. Zooplankton densities remained high throughout our experiments and were not influenced by perch or ruffe density (unpublished data). Large zooplankton was scarce and thus zooplankton may not have been a suitable alternative for the larger perch [37]. Indeed, Bergman and Greenberg [23] were able to detect a diet shift in only the first of two trials. They attributed this outcome to a general seasonal decline in food resources during their second trial.

The presence of ruffe or higher densities of ruffe resulted in a reduced stomach content mass for perch. Estimates of food consumption by ruffe and perch calculated from growth and temperature data (Tables 1 and 2) agree with the inferences made from stomach content mass (Figures 5 and 6). Perch daily ration and consumption declined in the presence of ruffe (ruffe and fish density experiment) and declined with increasing density of ruffe (gradient experiment). Several other studies on competition with perch have shown a shift in perch diet associated with competition, but no effect on the total amount of food consumed [23,29]; however, Dieterich et al. [38] found reduced consumption by European perch in the presence of ruffe at low prey density. In our experiments, available benthos was reduced by ruffe (unpublished data) and suitably sized alternative prey (zooplankton) were apparently not available, resulting in lower stomach contents and decreased growth of yellow perch.

Greater food consumption by ruffe could explain the reduced growth of perch if food availability was limiting. Ruffe food consumption was consistently higher than that of perch in the same mesocosms. In the first experiment, total food consumption in treatments containing half perch and half ruffe was 30% greater than total food consumption in the treatments containing perch alone. In the gradient experiment, total food consumption increased asymptotically with ruffe density and in both experiments, total consumption by ruffe was higher than perch in all but one treatment (16 perch and 8 ruffe in the gradient experiment) even when more perch were present. This pattern suggests that ruffe exert a greater demand on the benthic prey resource than perch (see also [21]), but that intraspecific competition among ruffe begins to limit overall demand as ruffe density increases. Both ruffe and perch daily rations decreased with ruffe density, but no changes in ration were found with perch density.

### 3.3. Competitive Mechanisms and their Implications

There are two common mechanisms by which one population can reduce the food consumption and growth of another [28]. The first mechanism is exploitative (consumptive) competition for food, whereby ruffe eat sufficient numbers of benthic organisms to reduce the quantity or quality of benthic prey available to perch. The second mechanism is interference (encounter) competition, whereby some aspect of ruffe behavior or a behavioral interaction between ruffe and perch decreases the foraging efficiency of perch. For example, if ruffe chase perch while perch are trying to feed, then ruffe decrease the time perch actually spent feeding. In this scenario, not only would the energy intake of perch decrease, but the energetic costs of responding to ruffe aggression may well be large enough to result in a biologically significant increase in energy expenditure. Differences in activity rate can result in substantial differences in growth rate of yellow perch and slower growth may be associated with higher activity rates [39].

The literature is inconclusive on the likelihood that ruffe interfere with perch. Dieterich et al. [38] inferred that interference rather than exploitation was the mechanism of competition between European perch and ruffe. Savino and Kolar [13] described ruffe as more aggressive than yellow perch, based on observations of behaviors such as the initiation of chases and prey stealing. Ruffe aggression was observed towards perch and among themselves [20]. Logically, the energetic cost of agonistic behavior would increase for both ruffe and perch as ruffe density increased. Interference competition would have the same result as resource competition insofar as growth is concerned. Savino and Kostich [20] suggested that ruffe intraspecific competition should be greatest at low and high ruffe densities and growth should be best at intermediate ruffe densities. In the gradient experiment, ruffe growth decreased linearly with increasing ruffe density; however, our densities were much lower than Savino and Kostich [20] used. The turbid waters of Perch Lake precluded any behavioral observations during the experiment.

The observed effects of ruffe on yellow perch growth and diet could be explained entirely by interference competition. However, in 1996, total abundance of benthic macroinvertebrates declined in the presence of ruffe and declines in Oligochaeta and Ceratopoginidae abundance and biomass and size of Chironomidae and Ceratopogonidae were related to presence of ruffe (unpublished data), suggesting consumptive (resource) competition. The decreasing abundance of benthic macroinvertebrates with increasing consumption by fish in the mesocosms reinforces this interpretation.

Bergman and Greenberg [23] did detect the suppression of several benthic taxa by ruffe, but the detection of effects of epibenthic predators on the benthic prey resource has proved elusive in other mesocosm studies and the factors that tend to obscure the link are extensively discussed by those authors (e.g., [30,40]). Our study suggests that consumptive competition partially explains the observed effect of ruffe on perch growth. However, because the growth pattern is consistent even when patterns of prey depletion are more complex, interference competition may be the more important mechanism.

Few studies have been able to definitively separate exploitative from interference competition directly. One approach for future studies would be to add an additional set of treatments whereby the competitor species would be allowed to feed alone for an extended period before being replaced by the other species. The effects of resource suppression by the first species would be apparent in the second species as consumptive competition without interspecific interference. The effects of interference competition could be obtained by subtracting the isolated (consumptive) effect from the competitive effect in sympatry.

The results of our experiments contribute to the understanding of the role of competition in the interactions between an introduced non-indigenous fish and the native fish community by clearly demonstrating competition between ruffe and yellow perch in a natural setting. The extent of harm done to economically important native fishes such as the yellow perch by invading ruffe will depend on a host of modifying environmental influences. For example, Brazner et al. [41] noted that ruffe avoided heavily vegetated littoral areas; ruffe may have less impact in vegetated and more complex systems. Also, Bronte et al. [4] indicated that observed declines in native fish abundance since the

introduction of ruffe were more likely the consequence of natural population dynamics rather than an effect of ruffe. However, they did find that in the case of yellow perch, ruffe were partially responsible for fluctuations in year class strength. Given the importance of North America's freshwater fisheries and the results reported here, the threat posed by ruffe and other exotic fishes to our aquatic ecosystems should be taken very seriously.

After their initial introduction to and expansion within the St. Louis River Estuary and Duluth Superior Harbor ruffe remained the most common fish found in assessment bottom trawls from 1990 through 2003 [35]. Since then, invasive round goby (*Neoogobius melanostomus*) have become abundant and alternated with ruffe as the most abundant fish between 2004 and 2011. Although ruffe are no longer the most abundant fish species in the St. Louis River, Duluth-Superior Harbor system, they remain quite abundant and have continued to increase in Chequamegon Bay [42]. Gunderson et al. [43] downplayed the impact of ruffe on native communities and Bronte et al. [4] found effects limited to yellow perch year class strength, however, the analysis of longer term (>20 yr) catch records in the St. Louis River Estuary and Duluth Superior Harbor indicates that native fish declined in the presence of ruffe after 1989 and did not begin to rebound until 20 years later [35]. Invasive round gobies are more likely to have negative impacts in hard bottom systems, whereas ruffe may be more impactful in soft bottom benthic communities [22,35] and low light habitats [11,17]. Efforts to control ruffe by enhancing native predators were largely ineffective because native predators prefer to consume native fish rather than ruffe [5], even when ruffe have been present for a number of years and are the most abundant prey [6,44]. Local reductions of ruffe may be accomplished with bottom trawling removal but physical removal is not an effective tool to control ruffe [45]; currently, there are no effective tools to selectively control ruffe.

Although ruffe have remained contained to the Great Lakes and have not spread beyond Lakes Superior, Michigan and Huron, there is considerable concern for their spread to Lake Erie and the Mississippi River Basin [7,25] and inland lakes and river systems [46] across the upper Midwest and northeastern North America [47]. Their expansion and success in inland lakes in Europe [6,48] indicates the potential for success in a variety of inland lakes. Stepien et al. [3] suggest that low genetic diversity might be limiting the success and expansion of ruffe beyond their current range in North America, however, it is likely that education and possession and transport restrictions have restricted the introduction of ruffe to inland waters. Once introduced to interconnected inland waters ruffe could become established in a number of systems and be of particular concern in lower light productive systems. The recent declines in water clarity in Lake Erie (e.g., [49]) could favor ruffe over yellow perch, particularly if macrophytes decline [16].

## 4. Materials and Methods

### 4.1. General Procedures

We conducted two sets of mesocosm experiments, the first in 1996 to assess the effect of ruffe (with and without) and total fish density (high and low) on yellow perch growth and diet and the second in 1997 to examine the effects of an increasing density of ruffe on perch and the effects of an increasing density of perch on ruffe. The mesocosm experiments were conducted in Perch Lake, a shallow backwater of the St. Louis River approximately 20-km upstream from Duluth Harbor (46°45′ N, 92°06′ W). Sixteen enclosures, arranged in four blocks of four units each, were established in the lake to house the experimental treatments. Enclosures were open-ended cylinders, approximately 1.7 m in height and 4 m in diameter, made of 12 mil polyethylene. The bottom margin of each cylinder, a weighted semi-rigid collar, was forced into the soft-muck sediments by SCUBA divers. Approximately 12.6 m$^2$ of benthic surface was enclosed. The cylinder walls were secured at the top to a ring of PVC pipe that was attached to a floating wooden platform. The platforms held the open ends of the mesocosms above the waterline and served as working platforms from which to access the mesocosms. Monofilament netting was suspended across the tops of the mesocosms when

unattended to prevent avian predation. Prior to the experiments, mesocosms were electrofished to remove fish trapped during installation. The dominant open sediment benthic macroinvertebrates were Oligochaeta, Chironomidae, Ceratopogonidae, and *Chaoborus*. The zooplankton community included *Bosmina*, *Chydorus*, *Daphnia*, *Ceriodaphnia*, and Calanoida and Cyclopoida Copepoda.

Ruffe were collected from the St. Louis River Duluth-Superior Harbor by bottom trawl. Depth, tow duration, and location varied between collections. Since sufficient numbers of suitable size perch could not be obtained from the St. Louis River, perch for the experiment were beach seined from Oak Lake near Duquette, MN. Perch and ruffe were selected to be as similar in length and mass as possible, but perch were about 5g heavier than ruffe in 1996 (15 g vs. 10 g) and 3g heavier in 1997 (11 g vs. 8 g). The average total length ± 95% confidence interval of ruffe used in the 1996 experiment was 99 ± 1.0 mm; perch were 116 ± 0.4 mm. In 1997, ruffe mean total length was 91 ± 0.7 mm and perch mean total length was 101 ± 1.5 mm. Perch and ruffe were held in 1.9 m$^3$ flow-through tanks, positioned in Perch Lake, for up to one week following capture to allow for mortality associated with capture and handling stress. Individuals of both species were anaesthetized with a 0.5 mL/L solution of 2-phenoxyethanol, measured (total length ± 1 mm), and individually marked with passive integrated transponder (PIT) tags. After marking, fish were held for up to one week in the flow-through tanks before stocking into the mesocosms. Fish were then anaesthetized, identified, weighed to the nearest 0.1 g, and assigned to treatments (mesocosms) in a stratified random manner. Mean fish mass at the beginning and end of each experiment are given in Tables S1–S4.

Perch and ruffe were recovered from the mesocosms at the midpoint and end of each experiment by angling. Previous experience with electrofishing and purse seines indicated that angling was the most effective means of retrieving fish. Furthermore, angling minimized disturbance to the substrate and other biota when sampling at the midpoint of an experiment. Electrofishing was conducted after angling at the end of each experiment to retrieve any remaining fish.

Captured fish were immediately anaesthetized in individual plastic tubs with a 120 mg/L solution of MS-222. Regurgitated prey were rarely seen in the MS-222 bath, but when found they were preserved along with other stomach contents. Since only a few seconds elapsed between hooking a fish and placing it into the anaesthetic bath, there was little opportunity for regurgitation during angling. Since we could not be so confident about stomach contents of fish captured by electrofishing, we excluded these fish from the diet analysis. Anaesthetized fish were identified, weighed, and measured. Fish captured during mid-experiment sampling had their stomachs flushed with water to remove contents before being returned to their mesocosms. Stomach contents were preserved in 80% ethanol.

At the conclusion of the experiment in 1996, fish were immediately euthanized with MS-222, sealed in plastic bags, and embedded in ice. Fish were later identified, weighed, and measured before being frozen. Frozen fish later were thawed and dissected in the laboratory, at which time stomachs were removed and PIT tags were recovered. In 1997, fish were weighed, measured, and dissected in the field as they were captured. Their stomachs and PIT tags were immediately placed in 80% ethanol. This change was made because an experiment replicating the 1996 handling procedures showed that a mass loss of approximately 3% of live mass occurred during temporary storage on ice. Masses reported for 1996 were adjusted to reflect live mass.

Stomach contents were examined in the laboratory with a dissecting microscope at a magnification of 10×. Prey items were identified based on [50] to a 9 taxonomic groups (Ephemeroptera, Trichoptera, *Chaoborus*, Ceratopogonidae, Chironomidae, Amphipoda, Ostracoda, Cladocera and Copepoda) chosen for feasibility of identification and because body-part, length-biomass (dry mass of intact specimens) relations were available in the literature [51–53]. Ingested bait was not counted as prey. Occasionally, prey items were found in fish stomachs, such as winged adult insects, that could not be readily classified into any of these categories. If, however, the unusual prey item or items were large enough to obtain a dry weight, then the mass was included with the other estimates. The total mass of stomach contents of each fish was calculated, and the proportion, by mass, of the diet composed of microcrustaceans was determined. Ostracoda, Copepoda, and Cladocera were considered microcrustaceans [29]. Although

Copepoda and Cladocera were not systematically identified to greater taxonomic detail, some of these organisms were probably benthic rather than planktonic. The diet composition of perch and ruffe are given in Tables S5 and S6.

### 4.2. Ruffe and Fish Density Experiment Design and Analysis

This experiment, conducted in 1996, used a factorial design (high and low density, with or without ruffe) to examine the effects of ruffe on perch growth and diet at two overall fish densities. Four treatments in each of four blocked replicates were established as follows: 28 perch, 14 perch + 14 ruffe, 14 perch, and 7 perch + 7 ruffe. To account for potential spatial variation in Perch Lake, mesocosms were arranged in blocks; each block contained a replicate of each treatment. Fish were added to the mesocosms on 12 September and the experiment was concluded after 6 weeks on the 24th of October. The mean biomass stocked into each treatment is given in Table 3. These densities are similar to those used in other ruffe and perch experiments (e.g., [23,40]) and were representative of localized densities that were seen in the St. Louis River Duluth Superior Harbor.

**Table 3.** Mean total biomass ± 95% C.I. of ruffe (R) and yellow perch (P) in treatments of the 1996 ruffe and fish density experiment. There were four replicates of each treatment.

| Treatments | High Perch Alone | High Perch and Ruffe | Low Perch Alone | Low Perch and Ruffe |
|---|---|---|---|---|
| Species Numbers | 0R 28P | 14R 14P | 0R 14P | 7R 7P |
| Ruffe (g) | | 152 ± 15 | | 74 ± 7 |
| Perch (g) | 435 ± 12 | 226 ± 30 | 229 ± 25 | 116 ± 10 |

At 3 and 5 weeks after stocking, samples of ruffe and perch were captured, weighed, and stomach contents preserved. Fish not returned at week 3 were replaced, but fish removed at week 5 were not. At the end of the experiment all fish captured were identified, weighed, and stomach contents preserved. Mean individual fish mass and total length at the beginning and end of the experiment are given in Tables S1 and S2 [54].

A repeated measures blocked split-plot ANCOVA [55] was used to test the effects of block, overall fish density (whole plot), presence or absence of ruffe (whole plot), time (sub-plot), and overall fish biomass (covariate), on the growth response of perch. Data from all three sampling periods were included. One replicate of perch and ruffe at low density was excluded from the analysis because of poor fish recovery and extremely high growth of the few fish recovered.

Instantaneous growth rate was used as the growth response:

$$IG = \ln(Wt_2/Wt_1) \tag{1}$$

where $IG$ = instantaneous growth rate, $Wt_2$ = final fish mass, and $Wt_1$ = initial fish mass.

Because there was little variation in the initial mass of the perch and all fish were in the experiment for the same amount of time, gross change in mass ($Wt_2 - Wt_1$) during the experiment was also analyzed as an alternative response. Replacement fish were not included in the analysis of growth.

Two measures of diet response were analyzed with a blocked split-plot ANCOVA: mean stomach content mass and the proportion (arcsin transformed), by mass, of microcrustaceans in the diet. Replacement fish were included in the analysis of diet response. Diet response was also analyzed for the final (24 October) sample only using a factorial ANCOVA that included block, fish density, biomass (covariate), and the presence or absence of ruffe.

*4.3. Ruffe and Perch Density Gradient Experiment Design and Analysis*

The intent of the 1997 experiments was to compare the effect of ruffe density on yellow perch growth and diet with the effect of perch density on ruffe growth and diet. To that end, two series of treatments were devised by superimposing an increasing density of one species (0, 4, 8, and 16 fish) on a constant density of the other (8 fish). There were two replicates of each treatment in each trial for a total of 14 experimental units (mesocosms). The experiment was executed twice, resulting in four replicates of each treatment. Although the mesocosms were arranged in groups of four, treatments were assigned randomly to the mesocosms. A block size of 7 mesocosms was not feasible given the practical limitations of construction and wind conditions on Perch Lake.

The duration of each trial was 5 weeks. At 2.5 weeks, samples of ruffe and perch were captured (by angling), identified, weighed, and stomach contents preserved. Fish not returned were replaced. The first trial commenced on the 11th of August and concluded on the 15th of September. The second trial commenced on the 22nd of September and concluded on the 28th of October. Mean biomass stocked into each treatment is given in Table 4. Mean fish mass at the beginning and end of each trial are given in Tables S3 and S4 [54].

**Table 4.** Mean total biomass ± 95% C.I. of ruffe (R) and yellow perch (P) in treatments of the 1997 fish density gradient experiment. There were four replicates of each treatment except for the 0R + 8P and 4R + 8P treatments for which there were three replicates and 8R + 8P (6 replicates).

| | Increasing Ruffe | | | |
|---|---|---|---|---|
| **Species Numbers** | **0R** **8P** | **4R** **8P** | **8R** **8P** | **16R** **8P** |
| Ruffe (g) | | 29 ± 18 | 60 ± 5 | 118 ± 18 |
| Perch (g) | 82 ± 50 | 64 ± 32 | 85 ± 25 | 87 ± 45 |
| | Increasing Perch | | | |
| **Species Numbers** | **0P** **8R** | **4P** **8R** | **8P** **8R** | **16P** **8R** |
| Perch (g) | | 53 ± 36 | 85 ± 25 | 179 ± 28 |
| Ruffe (g) | 58 ± 11 | 61 ± 11 | 60 ± 5 | 60 ± 12 |

The effect of adding ruffe to a constant density of perch on the growth response of both species was analyzed with a factorial ANOVA. The analysis included the effects of trial, species (ruffe or yellow perch) and number of ruffe present. A linear contrast was used to test the effect of the increasing density of ruffe.

Specific daily growth rate was used as the response in this experiment:

$$G = [\ln(Wt_2/Wt_1) * (t_2 - t_1) - 1] \times 100 \tag{2}$$

where $G$ = specific daily growth rate, $Wt_2$ = final fish mass, $Wt_1$ = initial fish mass, and $t_2 - t_1$ = interval (days) between measurements [56].

The specific daily growth rate was used because some mortality occurred early in the experiment necessitating the addition of replacement fish. Due to the smaller numbers of fish used in these experiments relative to the 1996 experiment, these replacements together with fish added at mid-experiment to replace angling mortalities constituted a substantial portion of the available sample and thus could not be excluded as in 1996.

The effect on fish growth response of adding perch to constant ruffe density was analyzed separately for each trial due to missing observations in the second trial. Three observations had to be discarded due to a stocking error in the second trial. A factorial ANOVA was used to analyze diet response (mean stomach content mass per treatment and arcsin transformed proportion, by mass, of microcrustaceans in the diet) of both species to increasing ruffe.

### 4.4. Bioenergetics Modeling

Food consumption in each mesocosm was estimated from water temperature (recorded continuously in 1996 and weekly in 1997), prey type, and growth of ruffe and perch during the experiments. Mean initial and final mass of ruffe and perch in each mesocosm were used to parameterize the models and estimate mean individual consumption. Since final mean mass was influenced by replacement fish in 1997, final mean individual mass was calculated from specific growth rate. Mean water temperature in 1996 was 13.1 °C. Mean water temperature in 1997 was 19.1 °C during the first trial and 11.8 °C during the second trial.

Food consumption of ruffe was estimated using a relationship between food consumption, temperature (*T*), and specific daily growth (*G*) derived from the laboratory results reported in Henson and Newman [21] because formal bioenergetics models are not available for ruffe (e.g., [57]). Daily ration (*DR*; % of live mass) was estimated via multiple regression as:

$$DR = 10.24\ G + 0.22\ T - 0.225\ (r^2 = 0.96, p < 0.001) \tag{3}$$

The ruffe in the mesocosms and the ruffe used in Henson and Newman [21] were of similar size (9 to 10 g) and consumed exclusively invertebrate prey. Mean ruffe mass was 10.8 g in 1996 and 7.4 g in 1997.

For yellow perch, food consumption was estimated using Fish Bioenergetics 3.0 [57]. This software uses bioenergetic relationships from Kitchell and Stewart [58] to predict yellow perch consumption given growth and temperature. The model provides good agreement with field estimates for age 1–3 yellow perch at water temperatures < 22 °C [59], conditions that were met by our data. Mean initial and final masses from each mesocosm were used, along with the temperature data and the fit *p*-value to estimate consumption. Prey energy density was entered as 2500 joules/g wet mass to represent the mixed invertebrate diet of the perch. Mean perch mass was 16.0 g in 1996 and 10.9 g in 1997. Mean (and SE) consumption for ruffe and perch was averaged over the replicate mesocosms in each treatment. The intent of the 1997 experiments was to compare the effect of ruffe density on yellow perch.

**Supplementary Materials:** The following are available online at http://www.mdpi.com/2410-3888/5/4/33/s1, Table S1: Initial and final mean individual live masses and mean individual total lengths of perch in the 1996 mesocosm experiment, Table S2: Initial and final mean individual live masses and mean individual total lengths of ruffe in the 1996 mesocosm experiment, Table S3: Initial and final mean individual live masses and mean individual total lengths of perch in the 1997 mesocosm experiment, Table S4: Initial and final mean individual live masses and mean individual total lengths of ruffe in the 1997 mesocosm experiment, Table S5: Diet composition of ruffe and perch at the conclusion of the 1996 mesocosm experiment, Table S6: Diet composition of ruffe and perch at the conclusion of the first trial of the 1997 mesocosm experiment.

**Author Contributions:** Conceptualization, R.M.N. and C.R.; methodology, F.G.H. and R.M.N.; formal analysis, F.G.H.; writing—original draft preparation, F.G.H. and R.M.N.; writing—review and editing, F.G.H. and R.M.N. and C.R.; visualization, F.G.H.; supervision, R.M.N.; project administration, R.M.N. and C.R.; funding acquisition, R.M.N. and C.R. All authors have read and agreed to the published version of the manuscript.

**Funding:** This work was prepared by the authors using federal funds under award USDOC-NA46RG0101 (C. Richards, PI) from Minnesota Sea Grant, National Sea Grant College Program, National Oceanic and Atmospheric Administration, U.S. Department of Commerce. The statements, findings, conclusions, and recommendations are those of the authors and do not necessarily reflect the views of NOAA, the Sea Grant College Program or the U.S. Department of Commerce. Additional support was provided by the Minnesota Agricultural Experiment Station under Project 74 and USDA National Institute of Food and Agriculture, Hatch grant MIN-41-081.

**Acknowledgments:** This manuscript was originally a chapter of Fred Henson's MS thesis that languished as we all moved on to other endeavors. Jeff Schuldt was instrumental in coordinating and overseeing these experiments. We thank Jeremy Trexel, Jim Gangl, Jeff Schuldt and the NRRI technical staff for their steadfast intellectual and physical labor which made this complex field experiment function. We are also grateful for the occasional assistance of Eric Merten, Geoff Schrag, Vanessa Pepi, and Jessica Gurley. Many thanks to Sanford Weisberg for statistical advice. The U.S. Government is authorized to reproduce and distribute reprints for government purposes, not withstanding any copyright notation that may appear hereon.

**Conflicts of Interest:** The authors declare no conflict of interest. The funders had no role in the design of the study; in the collection, analyses, or interpretation of data; in the writing of the manuscript, or in the decision to publish the results.

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
