# Peer review of "Competition between Invasive Ruffe (Gymnocephalus cernua) and Native Yellow Perch (Perca flavescens) in Experimental Mesocosms"

_fishes, doi:10.3390/fishes5040033_

Round 1
Reviewer 1 Report
This is a timely Ms addressing a very updated subject and reads very well. The Introduction provides a very good state-of-the-art to the topic to be investigated and is supported by adequate references. There is a good connection with the objectives of the study which are clearly addressed.
The methods are described in sufficient detail but may be some references indicated the bibliography used in the identification of stomach contents would be advisable (lines 448-440)
The Results are adequately and sufficiently described and are adequately discussed. There is a good integration with the conservation implications deriving from the negative impacts of invasive species
The References section needs some revising as, for example, some journals titles are presented in full, others not. Please revise to keep a consistent format throughout the list of references.
My suggestion is that the Ms might be accepted as soon as it is updated with the suggestions referred to above.
Author Response
This is a timely Ms addressing a very updated subject and reads very well. The Introduction provides a very good state-of-the-art to the topic to be investigated and is supported by adequate references. There is a good connection with the objectives of the study which are clearly addressed.
We are glad the reviewer found the manuscript timely and useful.
The methods are described in sufficient detail but may be some references indicated the bibliography used in the identification of stomach contents would be advisable (lines 448-440)
We added citation to the reference (Thorp and Covich) we used to indentify the invertebrates on line 442 (now ref 48).
The References section needs some revising as, for example, some journals titles are presented in full, others not. Please revise to keep a consistent format throughout the list of references.
We have gone through the references and corrected formatting to match journal style.
Reviewer 2 Report
Dear Academic Editor FISHES,
I attached to you the review of the manuscript: Newman et al.942025,
Titled: “Competition between invasive ruffe (Gymnocephalus cernua) and native yellow perch (Perca flavescens) in experimental mesocosms”.
General comments:
The above paper reports an analysis of experimental mesocosms on the influence of the alien ruffe (Gymnocephalus cernua) on the native yellow perch (Perca flavescens). My overall impression is that this is an interesting piece of research that addresses a pertinent topic in ecology, with repercussions in conservation and management.
The first thing that struck me is that the two experiments described in the study were conducted 14 years ago. I assume that this study was the result of a State effort to control allien ruffe. In this sense, I would like to know what measures have been taken in these 14 years based on the information reported, and if the report led to some type of action in nature.
Specific comment:
1) Why were the experiments carried out in September-October and August-September, respectively? Why did August start the following year?
2) In Lines 138-141 the authors compare the results of both experiments, how was it compared? According to the authors, the growth rates used were different. Furthermore, since the growth rates used were different, and the months and period of the study also differ, I do not know to what extent the two experiments are comparable. Although it is true that the results of both experiments separately point to the same effect of ruffa on yellow perch.
3) Line 119 what other factors? specify please
4) Bioenergetics modeling, it was adjusted to a linear regression, why?
5) Finally, I am concerned about a possible misinterpretation of the results of the first experiment. It is true that in the case of 0 Ruffe, and low density, the yellow perch has an instantaneous rate of positive growth, however, at the third week it is double that at the 5 week, with high variability, and finally the 6 weeks is negative. There could be a different effect here than Ruffe or interspecific competition, and I think it should be discussed, as it is the main weakness of the study.
Author Response
The first thing that struck me is that the two experiments described in the study were conducted 14 years ago. I assume that this study was the result of a State effort to control allien ruffe. In this sense, I would like to know what measures have been taken in these 14 years based on the information reported, and if the report led to some type of action in nature.
This work was independent of managment efforts and did not directly inform managment actions but we have included serveral sentences decribing actions taken and results of those action at lines 373-377. We decided that this information is better presented in the discussion rather than introduction.
1) Why were the experiments carried out in September-October and August-September, respectively? Why did August start the following year?
The first set of experiments were conducted later partly due to logistics of setting up the mesocosms and experiements during the first year. We were able to start earlier the second year and needed to do this to complete the second replicate by October that year. We could not start the experiments earlier than August because we needed to collect wild fish to stock in mesocosms at an appropriate size. It is not clear that this information enhances the manuscript but we could add it if the editor requests.
2) In Lines 138-141 the authors compare the results of both experiments, how was it compared? According to the authors, the growth rates used were different. Furthermore, since the growth rates used were different, and the months and period of the study also differ, I do not know to what extent the two experiments are comparable. Although it is true that the results of both experiments separately point to the same effect of ruffa on yellow perch.
In lines 138-141 we are comparing the two replicate blocks of the second experiment for increasing ruffe density so these are directly comparable with the use of specific growth rate. We do not attempt to explicilty compare growth across the two years, just the results of the statistical tests. For increasing perch density we do give separate results for each block (lines 145-148) as there were missing values in this set. The rationale and description of these differnces are given in the methods.
3) Line 119 what other factors? specify please
We beleive this must refer to line 121. We added the two other factors in the model that did not have interactions, time and overall fish density at line 121-122. Part of the reason for these questions is that the methods follow the results and discussion - the variables assess are clearly layed out in the methods.
4) Bioenergetics modeling, it was adjusted to a linear regression, why?
Food consumption for perch was estimated with the formal bioenergetics model of Hanson et al. That model has not be developed for ruffe so we used the described regression approach to estimate consumption by ruffe. We now indicate that a formal model was not available for ruffe on line 556.
5) Finally, I am concerned about a possible misinterpretation of the results of the first experiment. It is true that in the case of 0 Ruffe, and low density, the yellow perch has an instantaneous rate of positive growth, however, at the third week it is double that at the 5 week, with high variability, and finally the 6 weeks is negative. There could be a different effect here than Ruffe or interspecific competition, and I think it should be discussed, as it is the main weakness of the study.
We have added a brief discussion of these issues at lines 270 to 279.